# The Essential Oil Compositions of Three *Teucrium* Taxa Growing Wild in Sicily: HCA and PCA Analyses

**DOI:** 10.3390/molecules26030643

**Published:** 2021-01-26

**Authors:** Giorgia Catinella, Natale Badalamenti, Vincenzo Ilardi, Sergio Rosselli, Laura De Martino, Maurizio Bruno

**Affiliations:** 1Department of Food, Environmental and Nutritional Sciences (DeFENS), University of Milan, Via Celoria 2, 20133 Milan, Italy; giorgia.catinella@unimi.it; 2Department of Biological, Chemical and Pharmaceutical Sciences and Technologies (STEBICEF), University of Palermo, Viale delle Scienze, Parco d’Orleans II, 90128 Palermo, Italy; maurizio.bruno@unipa.it; 3Department of Earth and Marine Sciences (DISTeM), University of Palermo, Via Archirafi 26, 90133 Palermo, Italy; vincenzo.ilardi@unipa.it; 4Department of Agricultural and Forest Sciences (SAF), University of Palermo, Viale delle Scienze, Parco d’Orleans II, 90128 Palermo, Italy; sergio.rosselli@unipa.it; 5Centro Interdipartimentale di Ricerca “Riutilizzo Bio-Based Degli Scarti da Matrici Agroalimentari” (RIVIVE), University of Palermo, 90128 Palermo, Italy; 6Department of Pharmacy, University of Salerno, 84084 Fisciano, Italy

**Keywords:** *Teucrium montanum* L., *Teucrium capitatum* L., *Teucrium flavum* L., essential oils, GC-MS, PCA, chemotaxonomy

## Abstract

The chemical composition and the qualitative and quantitative variability of the essential oils of three taxa belonging to the *Teucrium* genus were studied. The investigated taxa, that grow wild in Sicily, were *Teucrium flavum* L. (section *Chamaedrys* (Mill.) Scheb.), *Teucrium montanum* and *Teucrium capitatum* L. of section *Polium* (Mill.) Scheb. Essential oils were extracted by hydrodistillation and analyzed by GC-MS. In total, 74 compounds were identified. Sesquiterpene hydrocarbons were found to be the main group for *T. flavum* (48.3%). *T. capitatum* consisted essentially of monoterpene hydrocarbons (72.7%), with *α*-pinene (19.9%), *β*-pinene (27.6%) and sylvestrene (16.6%) as the most abundant compounds whereas ledene oxide (12.1%), epiglobulol (13.5%) and longifolenaldehyde (14.5%) were identified as the main constituents among the oxygenated sesquiterpenes (63.5%) of *T. montanum*. Furthermore, a complete literature review on the composition of the essential oils of all the other accessions of these *Teucrium* taxa, studied so far, was performed. Hierarchical Cluster Analysis (HCA) and Principal Component Analyses (PCA) were used in order to demonstrate geographical variations in the composition of the essential oils.

## 1. Introduction

According to The Plant List [1] more than nine hundred and fifty scientific plant names of species rank for the genus *Teucrium* are present. Of these, more than three hundred are accepted names, including species, subspecies, varieties, forms and hybrids. The southern, south-western and south-eastern parts of Europe are considered as the main center of differentiation of the genus although a significant number of these species grow also in Central Asia, south-western Asia, north-western Africa, southern North America, south-western South America and Australia [2,3]. These perennial, bushy or herbaceous plants live commonly in sunny habitats [4] and on the basis of their calyx shape, inflorescence structure and pollen morphology they have been divided into ten sections (*Teucropsis* Benth., *Teucrium* Benth., *Chamaedrys* (Mill.) Schreb., *Polium* (Mill.) Schreb., *Isotriodon* Boiss., *Pycnobotrys* Benth., *Scorodonia* (Hill) Schreb., *Stachyobotrys* Benth., *Scordium* (Mill.) Benth. and *Spinularia* Boiss.) [4,5,6], although different authors modified this number from eight up to fifteen and created several subsections [7,8].

In Sicily, the genus *Teucrium* is represented by ten taxa: *Teucrium fruticans* L., *Teucrium campanulatum* L., *Teucrium siculum* (Raf.) Guss., *Teucrium scordium* subsp. *scordioides* (Schreb.) Arcang., *Teucrium spinosum* L., *Teucrium chamaedrys* L., *Teucrium flavum* L., *Teucrium montanum* L., *Teucrium luteum* (Mill.) Degen (syn. *Teucrium polium* L. subsp. *aureum* (Schreb.) Arcang.), and *Teucrium capitatum* L. (syn. *Teucrium polium* subsp. *capitatum* (L.) Arcang.) [9].

*Teucrium* species have been shown a rich source of *neo*-clerodane diterpenoids that, among the non-volatile constituents, are considered as chemotaxonomic markers of the genus. Up to now, 279 *neo*-clerodanes have been identified and their occurrence has been largely reviewed [10,11,12,13,14,15]. Other metabolites isolated from species of this genus include abietane diterpenes, sesquiterpenes, triterpenes, steroids, flavonoids, iridoids and aromatic compounds [15].

Since ancient times, species of *Teucrium* have been largely utilized, in traditional medicine, for their biological properties, including antimicrobial, anti-inflammatory, antispasmodic, insecticidal, anti-malaria, etc. and two complete reviews of all the traditional uses of *Teucrium* taxa have been recently published [16,17]. In the same reviews [16,18], a comprehensive survey of the chemical composition and biological properties of the essential oils isolated from *Teucrium* taxa was also provided.

*Teucrium flavum* L., according to the Raunkiær system [19], belongs to fruticose chamaephytes with erect and very branchy woody stems, often with a violet–purple color, covered by a thick patented or slightly reflected hairiness, of 0.5 mm. The leaves are velvety and on the upper side glossy, dark green in color. The flowers are hermaphroditic on about 1-cm peduncles with 10 ribs and 5 teeth halfway along the tube. It is a typical species of the Mediterranean Region (Steno-Medit.), quite widespread in Sicily and it is found commonly from sea level up to over 1000 m of altitude. It participates in the constitution of low garrigues and Mediterranean shrubs on rocky substrates, rubble, debris fans and stony slopes, preferably of a carbonate nature. The analyzed samples were collected, at about 380 m of altitude, within an area of about 1000 square meters of discontinuous shrub vegetation, with evident signs of previous fires, physiognomized by *Bupleurum fruticosum* L., *Pinus halepensis* Mill., *Pistacia lentiscus* L., *Arbutus unedo* L., *Chamaerops humilis* L., *Thymbra capitata* (L.) Cav., *Teucrium luteum* (Mill.) Degen and *T. fruticans* L. [9].

*T. montanum* L., is a species of about 10 cm, bushy or pulviniform with robust stems, ascending, sometimes decumbent; the young ones are often prostrate, whitish and tomentose. The leaves are lanceolate and acuminate at the apex with sparsely hairy green upper lamina. The flowers with about 1-mm pedicels are robust and pubescent with a white–cream or yellowish corolla of 12–13 mm. This species is present in South and Central Europe, up to Germany and Poland, Turkey and Algeria [9]. This species was collected in an environment of mountain garrigue on Dolomite substrates, between 1450 and 1550 m of altitude.

*T. capitatum* has been treated for a long time to the rank of subspecies of *Teucrium polium* L. (*T. polium* subsp. *capitatum*). Reassessed over time to a specific rank, it is characterized by emanating a sweet, pleasant odor; it has suffrutics of 8–25 cm densely tomentose, with creeping woody stems and erect branches divided in the inflorescence. The leaves are lanceolate with 3–4 teeth per side and in summer they are revolute. The flowers are gathered in closely spaced verticils to form capituliform inflorescences. *T. capitatum* has a glaucous color, white or reddish–white corolla and a dense silvery white garment [9]. The species was collected in a chalky garrigue, about 700 m above sea level, dominated by the presence of *Thymbra capitata* (L.) Cav. and often in association with *Foeniculum vulgare* (Mill.), *Euphorbia rigida* M. Bieb., *Plantago albicans* L., *Gypsofila arrostoi* Guss., *Hyparrhenia hirta* (L.) Stapf, *Diplotaxis harra* subsp. *crassifolia* (Raf.) Maire, etc. *T. capitatum* grows in all the southern parts of Europe, Middle East up to the Caspian Sea and North Africa.

The chemical compositions of the essential oils of three Sicilian accessions *Teucrium*, namely *T. siculum*, *T. fruticans*, and *T. scordium* subsp. *scordioides* have been recently published [20]. Consequently, in the frame of our ongoing research on Sicilian plants [21,22,23,24], and in order to improve the knowledge on genus *Teucrium*, we decided to investigate the chemical compositions of the other three Sicilian taxa of *Teucrium*, which have never been analyzed: *T. flavum*, belonging to Section *Chamaedrys*, and *T. montanum* and *T. capitatum*, belonging to Section *Polium*. We also screened the reported literature in order to find data concerning the chemical composition of their essential oils and we performed Hierarchical Cluster Analysis (HCA) and Principal Component Analyses (PCA) in order to find a similarity among the Sicilian accessions and the other taxa belonging the same species studied so far.

## 2. Results and Discussion

### 2.1. Composition of the Essential Oils

Hydrodistillation of the aerial parts of *T. flavum* (***O*(*f*)**) gave a pale-yellow oil. Overall, forty-one compounds were identified, representing 91.2% of the total compositions. The components are listed in Table 1 according to their retention indices on a DB-5 column and are classified on the basis of their chemical structures into five classes. This essential oil was rich in sesquiterpene hydrocarbons (48.3%). *β*-Bisabolene (26.8%) was, by far, the main components of this class as well as of the oil, followed by *β*-caryophyllene (6.6%), *γ*-cadinene (5.5%) and *α*-caryophyllene (3.1%). Oxygenated sesquiterpenes were present in a lower amount (11.0%) with caryophyllene oxide (3.1%) as the main compound of the class whereas oxygenated monoterpenes were practically absent. Monoterpene hydrocarbons accounted for 27.0%, limonene (12.7%) being the main product, followed by *α*-pinene and *β*-pinene (7.0% and 5.4%, respectively).

Table 2 reports the main components of the essential oils, obtained by hydrodistillation, of the other populations of *T. flavum*, previously investigated, as well as of the other accessions of *T. montanum* and *T. capitatum* collected in different countries.

A comparison of our data with those reported in the literature (Table 2) shows some very interesting points. For our comparison and further statistical analysis, we reported all the papers related only to essential oils achieved by hydrodistillation and only obtained from aerial parts. With the exception of the populations of Corsica (France) [35], Tuscany (Italy) and Liguria (Italy) [48], all the other accessions indicate sesquiterpene hydrocarbons as the main class of the oils. Among these ones, the plants collected in Montenegro [27] showed a very similar profile with respect to (***O***(***f***)). In fact, in this oil, the main sesquiterpene components were *β*-bisabolene (35.0%) and *β*-caryophyllene (5.4%) and among the monoterpene hydrocarbons, as for (***O***(***f***)), the principal metabolites were *α*-pinene (17.5%), *β*-pinene (11.5%) and limonene (6.4%). A study on the different vegetative parts of a Sicilian population of *T. flavum* has been published some years ago [51] but the results were not inserted in Table 2 since the oils were obtained with a different method (microwave-assisted hydrodistillation). By the way, also in this case, the main component of all the investigated parts was *β*-bisabolene, although, the second more abundant compound, germacrene D, was totally absent in (***O***(***f***)).

The oil of *T. capitatum* (***V***(***c***)) (twenty-five compounds) was characterized by a huge amount of monoterpene hydrocarbons (72.7%), with *β*-pinene (27.6%), *α*-pinene (19.9%), sylvestrene (16.6%) and myrcene (8.6%) as main components. Oxygenated monoterpenes were present in a lower amount (12.2%) with *trans*-pinocarveol (3.1%) and terpinen-4-ol (2.7%) being the principal constituents of the class whereas sesquiterpene derivatives accounted only for 5.1%. Comparison with the data reported in Table 2 showed a good similarity with the population collected in Bulgaria [33]: *β*-pinene (26.8%), *α*-pinene (9.3%), although germacrene D, occurring in a good amount (17.7%) in this accession, was totally absent in (***V***(***c***)). Moreover, the population collected in Corsica [34,35,36] showed a high amount of pinene derivatives, although the ratio between *β*-pinene and *α*-pinene was reversed with respect to (***V***(***c***)). The absence is noteworthy in all the other oils of *T. capitatum*, studied so far, of sylvestrene.

The oil of *T. montanum* (***F***(***m***)) showed to be extremely rich in sesquiterpenes (94.3%) monoterpenes being only 0.9% of the total. Among oxygenated sesquiterpenes, the main class of the oil (63.5%), longifolenaldehyde (14.5%), epiglobulol (13.5%) and ledene oxide (12.1%) was the principal metabolites, whereas *β*-cedrene (8.9%) was the main constituent among the sesquiterpene hydrocarbons (30.8%). A comparison with the other accessions of *T. montanum* showed a peculiar profile of (***F***(***m***)). In fact, only the population from Jadovnik, Serbia [28,29], showed a poor amount of monoterpenes (7.6%), although it was very rich in sesquiterpene hydrocarbons (74.2%) rather than in oxygenated sesquiterpenes (7.0%). Furthermore, none of the main components of (***F***(***m***)) (longifolenaldehyde, epiglobulol and ledene oxide) were present in all the other accessions.

### 2.2. PCA and HCA Analyses of the Essential Oil Composition of Teucrium Taxa

As stated before, the main compounds of the essential oils of Teucrium taxa, collected in different accessions, and their relative abundance are reported in Table 2. For the compilation of the Table 2, the following points were considered: (I) only compounds with abundance ≥3%, and (II) only essential oils EOs obtained by hydrodistillation were taken into consideration.

The analyses were carried out considering the classes’ compounds with a significant contribution, according to the loading plot obtained by principal component analysis (PCA) for monoterpene hydrocarbons (MH), oxygenated monoterpenes (OM), sesquiterpene hydrocarbons (SH), oxygenated sesquiterpenes (OS) and other compounds (O).

For the *T. montanum* essential oils, as shown in the loading graph (Figure 1), all variables affected PC1 and PC2. In fact, PC1 (51.6%) was represented mainly by oxygenated sesquiterpene (OS) in the positive score, and in a minor contribution by MH, OM and O in negative scores; meanwhile, PC2 (27.7%) was represented mainly by a negative score of SH and positive scores of MH and OS.

HCA based on the Euclidean distance between groups indicated two species groups (A and B, Figure 2) identified by their essential oil chemotypes with a similarity <2.

A first Group A, in HCA analysis, the samples of *T. montanum* from Jabura (***E***(***m***)) and Mt. Orjen (***C***(***m***)), Serbia, whose oxygenated sesquiterpene compounds content (24.1–33.4%) differentiated them from the other oils were included. This cluster was characterized by the highest content of germacrene D (15.0%) and a relatively high content of *β*-caryophyllene (5.1–6.9%), *δ*-cadinene (3.6–4.5%) and *γ*-cadinene (3.6–4.1%). A definite Group B containing the samples collected in Croatia (***A***(***m***)) and Turkey (***B***(***m***)), was characterized by oils with a large percentage of monoterpene and sesquiterpene hydrocarbons; the difference between the two oils was mainly due to the O class (8.8–22.5%), which was the majority in Croatian EO.

The others two oils, Serbia Jadovnik (***D***(***m***)) and Sicily (***F***(***m***)), for their high contents of SH (74.2%) and OS (63.5%), respectively, were considered as two separate classes, without any similarity with the other clusters.

The PCA of *T. flavum* EOs (Figure 3), belonging to section *Chamaedrys*, presented a total variance of 90.2% of the original data. The most of samples showed a cluster formation in this model, affirming a similarity in the chemical composition of these essential oils. The PCA horizontal axis explained 55.3% of the total variance while the vertical axis a further 34.9%. HCA based on the Euclidean distance between groups indicated a solution with three clusters (A’, B’ and C’), with a dissimilarity <2 (Figure 4) which was mainly due to the variation along the major axis in PCA analysis. These clusters formed separate groups in the PCA biplot (Figure 3).

The first group (Group A’) representing the oils collected in Dalmatia (***A***(***f***)), Split (***B***(***f***)), River Moraca (***E***(***f***)), Liguria (***H***(***f***)) and Tuscany (***I***(***f***)), contained mainly monoterpene and sesquiterpene hydrocarbons (MH, SH) and had *α*-pinene, *β*-caryophyllene, *β*-pinene, germacrene D, limonene and *β*-bisabolene as the major compounds.

Different from this group, due to a variation in the positive scores of PC2 (34.9%), was the oil from Corsica (***M***(***f***)), essentially constituted by MH and characterized principally by *α*-pinene, *β*-pinene and limonene. Moreover, the ***M***(***f***) essential oil was out of the 95% confidence marked by the circle.

The second subgroup B’, instead, including *T. flavum* collected in Sicily (***O***(***f***)) and in Tunisia (***N***(***f***)), showed oils containing relevant quantities of both hydrocarbon and oxygenated sesquiterpenes (SH, OS). A marked difference was observed for Iranian oil (***D***(***f***)), which was abundant in SH (82.2%).

A third group (Group C’) was composed and the oils from Greece (***C***(***f***); ***L***(***f***)) and Marche (***F***(***f***); ***G***(***f***)); this group was rich in sesquiterpene hydrocarbons (SH, 39.0–49.4%) and others (O, 12.5%–26.8%).

With a dissimilarity <1, group C’ was divided in two subgroups (C’1 and C’2) in both HCA and PCA analysis, and was affected essentially by the variation along the vertical axis. Subgroup C’1 was represented by Greece EOs (***C***(***f***); ***L***(***f***)), containing a relatively high content of caryophyllene (12.2–13.5%), caryophyllene oxide (7.9–8.5%) and *α*-humulene (5.0–6.0%). Marche (***F***(***f***); ***G***(***f***)) EOs constituted the subgroup C’2. They are characterized by a moderate content of (*Z*,*E*)-*α*-farnesene (14.9–11.5%) and (*E*)-*β*-farnesene (5.7–7.3%) and a lower *β*-caryophyllene content (5.1–5.7%) than the B’1 subgroup.

The PCA of *T. capitatum* revealed that the first principal component (PC1 and PC2) represented the 60% of the total information (Figure 5). The general structure of the dendrogram (Figure 6) generated by HCA indicated the existence of three main clusters of populations, based on their chemical composition and the Euclidean distance between groups (distance < 2). The graph, in fact, presented a first large cluster (Group A”) formed by essential oils harvested in Boussaada (***A***(***c***)) (Algeria), Bouira (***B***(***c***)) (Algeria), Bulgaria (***C***(***c***)), Athens (***G***(***c***)), (Fonte Coberta (***N***(***c***)), Rabaça (***O***(***c***)), Cantanhede (***P***(***c***); ***Q***(***c***)), Serra D′Aire (***R***(***c***)), Portugal), Gennargentu (Sardinia) (***T***(***c***)) and Serbia (***U***(***c***)), which was characterized by a variable composition of monoterpenes and sesquiterpenes, both hydrocarbons and oxygenated ones.

With a dissimilarity <1, group A” was divided in three subgroups (A”1, A”2 and A”3) in HCA analysis. The first subgroup A”1 is formed by (***B***(***c***)) and (***T***(***c***)) essential oils, and it is marked by a high content of sesquiterpene hydrocarbons (37.8–43.7%), a medium level of oxygenated sequiterpenes (10.0–13.5%) and a null contribution of others (0–0.1%).

Subgroup A”2 was represented by (***C***(***c***)) and (***O***(***c***)) essential oils and it was characterized by the highest levels of *β*-pinene and germacrene D (26.8% and 17.7%, respectively). It also contains relatively higher amounts of sabinene (11.2%) and *α*-pinene (9.3%).

Finally, subgroup A”3 represented by (***A***(***c***)), (***Q***(***c***)) and (***R***(***c***)) samples, was characterized by a similar percentage of sesquiterpene hydrocarbons (31.2–33.2%), oxygenated sesquiterpenes (20.4–25.5%) and others (0–1.0%). (***A***(***c***)) essential oil was characterized by a higher content of *τ*-cadinol (18.3%), germacrene D (15.3%), *β*-pinene (10.5%), while Portugal essential oils (***Q***(***c***)) and (***R***(***c***)) was richer in terms of *δ*-cadinene (7.5–9.8%), *β*-caryophyllene (5.4–4.6%), *α*-cadinol (4.2–4.6%), respectively. However, the common point of these oils was the low or the completely absence percentage of the others class (O, ≤1.2%).

Another representative cluster (Group B”), characterized by the negative scores of PC1 (34.7%), was the one constituted by the oils from (Bustanico (***D***(***c***)), Corte (***E***(***c***)), Corsica), Porticciolo (***S***(***c***)) (Sardinia), Sicily (***V***(***c***)), abundant of monoterpene hydrocarbons (MH, 56.2–72.7%).

Lowering the level of dissimilarity (cut-off = 1, Figure 6), it is possible to divide this group into two subclusters (B”1 and B”2). Subcluster B”2 included two samples, (***D***(***c***)) and (***E***(***c***)), geographically close to each other and characterized by a high percentage of *α*-pinene (24.1–28.8%, respectively). The other two oils ((***S***(***c***)) and (***V***(***c***)), formed subgroup B”1. This cluster, in contrast to the previous subgroup B”2, had a higher content of hydrocarbon monoterpenes (62.8–72.7%) but a lower contribution of oxygenated monoterpenes (11.4–12.2%).

A third small group (C”) was represented by Crete oils (***F***(***c***); ***I***(***c***)), having a similar profile, which cannot be combined with Kos (***H***(***c***)) oil due to the enormous difference in the chemical composition of the classes. The Iran (***L***(***c***)) and Morocco (***M***(***c***)) oils influenced, respectively, the negative and positive scores of PC2 (25.3%): the first oil, the only one outside the confidence interval, was abundant in oxygenated sesquiterpenes (OS), in particular *α*-cadinol (46.2%) and caryophyllene oxide (25.9%), while the second one was rich of oxygenated monoterpenes (OM), such as *endo*-borneol (33.0%), bornyl acetate (15.6%), *α*-terpineol (12%) and *α*-thujol (10.9%).

## 3. Materials and Methods

### 3.1. Plant Material

Aerial parts of *T. flavum* (***O***(***f***)) were collected in June 2020, near Noto Antica (SR), (Sicily, Italy), at 380 m of altitude (36°57′25.37″ N and 15°02′18.76″ E). A voucher of the population analyzed was deposited in the herbarium of the University of Palermo (PAL 109709).

Aerial parts of *T. montanum* (***F***(***m***)) were sampled, in June 2020, in Contrada Quacella on the Madonie Mountains (Sicily, Italy) in an environment of mountain garrigue on Dolomite substrates, between 1450 and 1550 m of altitude in an area whose center had coordinates 37°50′49.78″ N; 14°1′22.59″ E. A voucher of the population analyzed was deposited in the herbarium of the University of Palermo (PAL 109708).

The aerial parts of *T. capitatum* (***V***(***c***)) were taken, in June 2020, in the chalky hilly center of Sicily, in an area in the countryside of Marianopoli (CL), Sicily, Italy with geographical coordinates 37°36′4.58″ N; 13°57′46.10″ E, about 700 m above sea level and the respective voucher was deposited in the herbarium of the University of Palermo (PAL 109711).

### 3.2. Essential Oil Extraction

A variable quantity of the aerial parts of *T. flavum*, *T. montanum* and *T. capitatum* (33–105 g) were subjected to hydrodistillation for 3 h second using Clevenger’s apparatus [52]. The oils (yields 0.08% (*v*/*w*), 0.15% and 0.07% for (***O***(***f***)), (***F***(***m***)) and (***V***(***c***)), respectively), were dried with anhydrous sodium sulphate, filtered and stored in the freezer at −20 °C, until the time of analysis.

### 3.3. Chemical Analysis of Essential Oils

Analysis of essential oil was performed according to the procedure reported by Rigano et al. [22]. EO analysis was performed using an Agilent 7000C GC (Agilent Technologies, Inc., Santa Clara, CA, USA) system, fitted with a fused silica Agilent DB-5 capillary column (30 m × 0.25 mm i.d.; 0.25 μm film thickness), coupled to an Agilent triple quadrupole Mass Selective Detector MSD 5973 (Agilent Technologies, Inc., Santa Clara, CA, USA); ionization voltage 70 eV; electron multiplier energy 2000 V; transfer line temperature, 295 °C. Solvent Delay: 4 min. The other GC analysis was performed with a Shimadzu QP 2010 plus equipped with an AOC-20i autoinjector (Shimadzu, Kyoto, Japan) gas chromatograph equipped with a Flame Ionization Detector (FID), a capillary column (HP-Innowax) 30 m × 0.25 mm i.d., film thickness 0.25 μm and a data processor. The oven program was as follows: temperature increase at 40 °C for 5 min, at a rate of 2 °C/min up to 260 °C, then isothermal for 20 min. Helium was used as the carrier gas (1 mL min^−1^). The injector and detector temperatures were set at 250 °C and 290 °C, respectively. An amount of 1 μL of each oil solution (3% EO/Hexane *v*/*v*) was injected with split mode. Linear retention indices (LRI) were determined by using retention times of *n*-alkanes (C_8_-C_40_) and the peaks were identified by comparison with mass spectra and by comparison of their relative retention indices with WILEY275 (Wiley), NIST 17 (NIST, The National Institute of Standards and Technology, Gaithersburg, MD, USA), ADAMS (Allured Business Media, Carol Stream, IL, USA) and FFNSC2 (Shimadzu, Kyoto, Japan) libraries.

### 3.4. Statistical Analysis

The essential-oil contents that exceeded 3.0% of the total oil composition in at least one species were considered as original variables and subjected, after normalization, to cluster analysis (CA) and to Principal component analysis (PCA). The statistical analyses were performed using PRIMER 6 (Massey University Eastbourne, Albany, New Zealand) with two principal components (PC) variables and the number of clusters were determined by using the rescaled distances in the dendrogram, using a cut-off point (Euclidean distance = 2) that allows the attainment of consistent clusters. The Principal Components Analysis (PCA) and the Hierarchical Cluster Analysis (HCA) were used to comprehend the similarity among the essential oils in relation to the contents of their chemical constituents. We tested two different cut-off similarity levels (cut-off level 1 and cut-off level 2), chosen on the basis of the mean distance between clusters measure and based on the similarities–differences between the samples belonging to the same cluster. Since the HCA analysis is a function of variables and observations, the highest correspondence between PCA and HCA resulted when we applied a cut-off of 2. A cut-off of 1 greatly increases the diversity between the analyzed theses and would lead to an incorrect clustering. The statistical analysis of the absence/presence was carried out using the cluster method of the PRIMER 6 software (Massey University Eastbourne, Albany, New Zealand) [53].

## 4. Conclusions

Chemical and statistical analyses (PCA and HCA) carried out on three Sicilian taxa (*T. flavum*, *T. montanum* and *T. capitatum*) belonging to the genus *Teucrium*, can provide chemotaxonomic information on the taxa investigated. All EOs were analyzed by GC-MS. *T. flavum* is mainly represented by sesquiterpene hydrocarbons (48.3%), the main class also found in most of all accessions already investigated. Sicilian *T. capitatum* essentially consisted of monoterpenes such as *α*-pinene (19.9%), *β*-pinene (27.6%) and sylvestrene (16.6%), finding a high similarity with the population collected in Bulgaria. *T. montanum*, on the other hand, showed a high abundance in both hydrocarbon and oxygenated sesquiterpenes (94.3%). The PCA analyses, and the subsequent analyses of the Clusters, based on the different chemical classes, represent a useful tool towards a complete taxonomic investigation, also leading to an understanding of the diversification of the genus *Teucrium.*

## Figures and Tables

**Figure 1 molecules-26-00643-f001:**
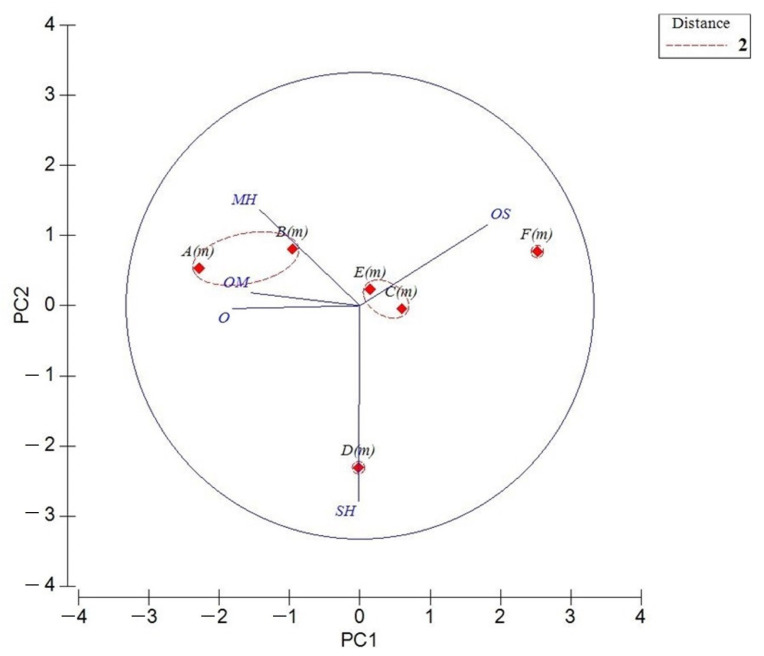
Principal component analysis (PCA) of the essential-oil composition of published *T. montanum* based on the principal classes of compounds: monoterpene hydrocarbons (MH), oxygenated monoterpenes (MO), sesquiterpenes hydrocarbons (SH), and oxygenated sesquiterpenes (OS) and others (O). The vectors shown are the eigenvectors of the covariance matrix. The plants’ codes are reported in Table 2.

**Figure 2 molecules-26-00643-f002:**
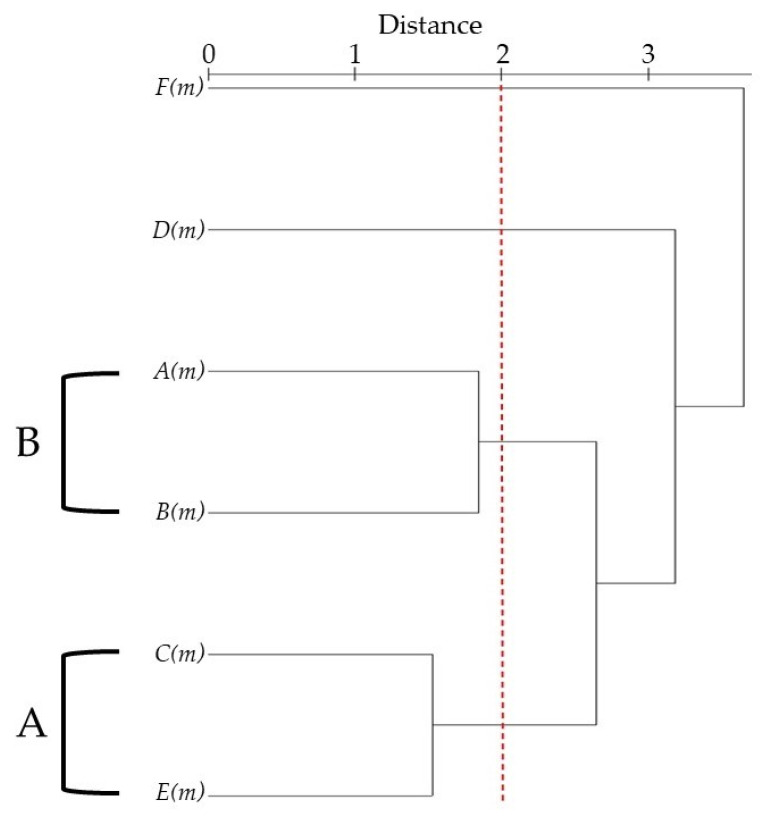
Dendrogram obtained by Hierarchical Cluster Analysis (HCA) based on the Euclidian distances between groups of ***A***(***m***), ***B***(***m***), ***C***(***m***), ***D***(***m***), ***E***(***m***) and ***F***(***m***) *Teucrium montanum* essential oils. The plants codes are reported in Table 2.

**Figure 3 molecules-26-00643-f003:**
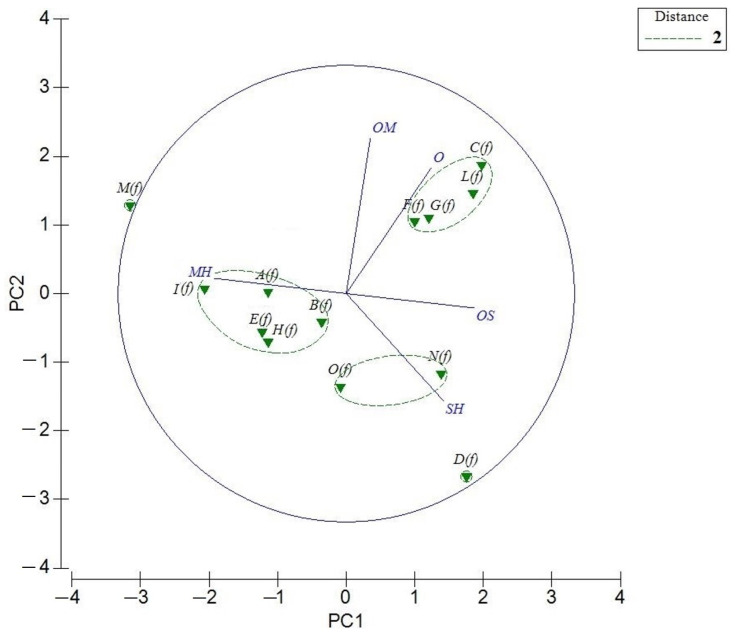
Principal component analysis (PCA) of the essential-oil composition of published *T. flavum* based on the principal classes of compounds: monoterpene hydrocarbons (MH), oxygenated monoterpenes (MO), sesquiterpenes hydrocarbons (SH), and oxygenated sesquiterpenes (OS) and others (O). The vectors shown are the eigenvectors of the covariance matrix. The plants codes are reported in Table 2.

**Figure 4 molecules-26-00643-f004:**
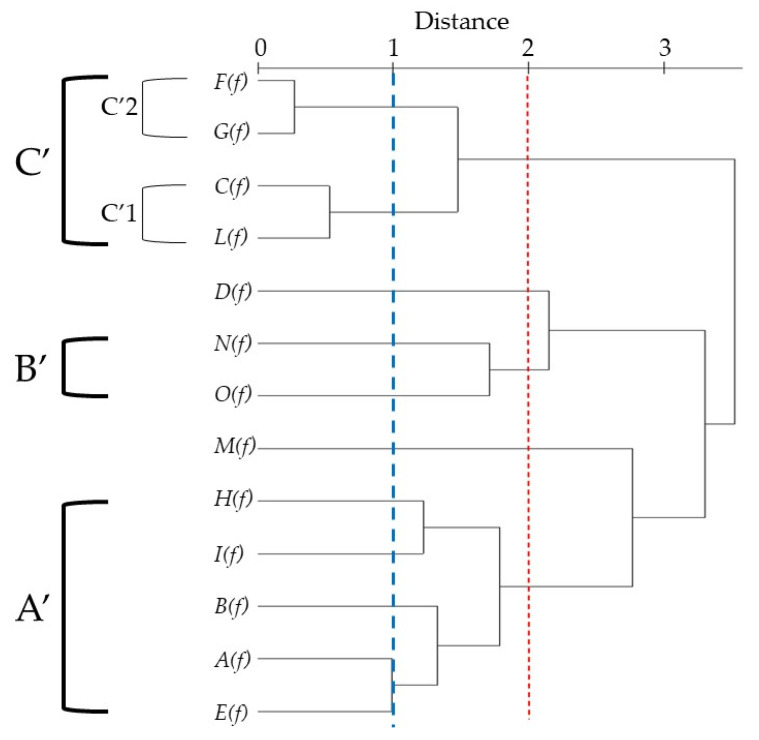
Dendrogram obtained by HCA based on the Euclidian distances between groups of ***A***(***f***), *B*(***f***), ***C***(***f***), ***D***(***f***), ***E***(***f***), ***F***(***f***), ***G***(***f***), ***H***(***f***), ***I***(***f***), ***L***(***f***), ***M***(***f***), ***N***(***f***) and ***O***(***f***) *Teucrium flavum* essential oils. The plants codes are reported in Table 2.

**Figure 5 molecules-26-00643-f005:**
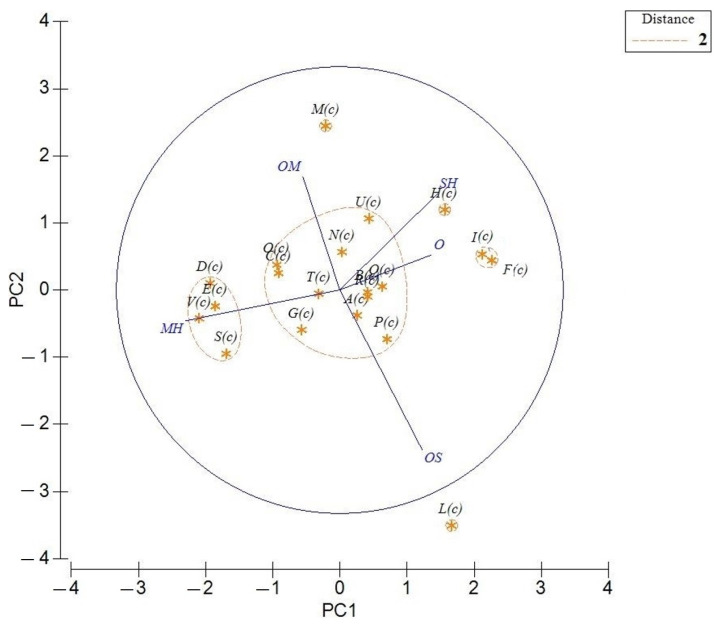
Principal component analysis (PCA) of the essential-oil composition of published *T. capitatum* based on the principal classes of compounds: monoterpene hydrocarbons (MH), oxygenated monoterpenes (MO), sesquiterpenes hydrocarbons (SH), and oxygenated sesquiterpenes (OS) and others (O). The vectors shown are the eigenvectors of the covariance matrix. The plants codes are reported in Table 2.

**Figure 6 molecules-26-00643-f006:**
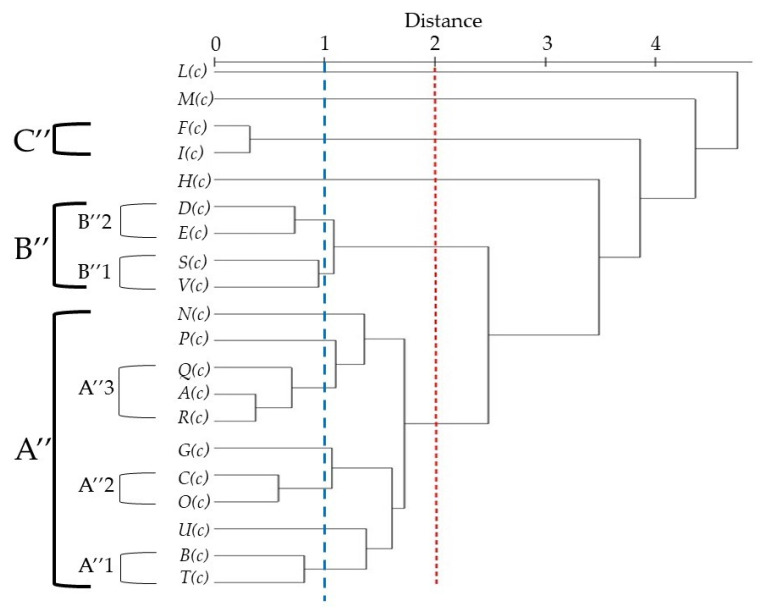
Dendrogram obtained by HCA based on the Euclidian distances between groups of ***A***(***c***), ***B***(***c***), ***C***(***c***), ***D***(***c***), ***E***(***c***), ***F***(***c***), ***G***(***c***), ***H***(***c***), ***I***(***c***), ***L***(***c***), ***M***(***c***), ***N***(***c***), ***O***(***c***), ***P***(***c***), ***Q***(***c***), ***R***(***c***), ***S***(***c***), ***T***(***c***), ***U***(***c***) and ***V***(***c***) *Teucrium capitatum* essential oils. The plants codes are reported in Table 2.

**Table 1 molecules-26-00643-t001:** Composition (%) of the essential oils of *T. flavum* (***O***(***f***)), *T. capitatum* (***V***(***c***)) and *T. montanum* (***F***(***m***)) collected in Sicily.

	Components Abbreviations	LRI ^a^	LRI ^b^	*T. flavum**O*(*f*)	*T. capitatum**V*(*c*)	*T. montanum**F*(*m*)
**1**	1-Hexen-3-ol	764	1244	-	0.1	-
**2**	*α*-Pinene ^c^	937	1026	7.0	19.9	0.4
**3**	Benzaldehyde ^c^	960	1478	-	1.9	-
**4**	*β*-Pinene ^c^	974	1105	5.4	27.6	-
**5**	Sabinene ^c^	976	1109	-	-	0.5
**6**	1-Octen-3-ol	983	1447	0.1	1.0	-
**7**	Myrcene ^c^	990	1152	0.2	8.6	-
**8**	Sylvestrene	1005	1200	-	16.6	-
**9**	Limonene ^c^	1027	1199	12.7	-	-
**10**	*cis*-Ocimene	1038	1233	0.2	-	-
**11**	*trans*-Ocimene	1050	1256	1.0	-	-
**12**	Linalool ^c^	1098	1546	0.3	-	-
**13**	Nonanal ^c^	1101	1380	0.2	-	0.3
**14**	Perillene	1105	1429	-	0.1	-
**15**	3-Octyl acetate	1118	1338	0.3	-	-
**16**	(*E*,*Z*)-2,6-Dimethyl-2,4,6-octatriene	1127	1345	0.5	-	-
**17**	*trans*-Pinocarveol	1132	1623	-	3.1	-
**18**	*δ*-Verbenol	1145	1652	-	1.1	-
**19**	Isobutyl caproate	1152	1333	0.2	-	-
**20**	2-Methylbutyl valerate	1159	1324	0.6	-	-
**21**	Terpinen-4-ol	1176	1586	-	2.7	-
**22**	*p*-Cymen-8-ol	1184	1822	-	0.4	-
**23**	Myrtenal	1190	1627	-	1.1	-
**24**	(*E*)-1-Octenyl acetate	1193	1376	1.0	-	-
**25**	Terpineol	1197	1710	-	1.6	-
**26**	Carvone ^c^	1199	1728	-	1.0	-
**27**	Myrtenol	1201	1751	-	0.7	-
**28**	3-Isopropylbenzaldehyde	1204	1456	-	0.2	-
**29**	*cis*-Carveol	1231	1780	-	0.2	-
**30**	Isoamyl caproate	1239	1449	1.7	-	-
**31**	Copaene	1378	1474	0.3	-	2.3
**32**	Isoledene	1380	1489	1.7	-	-
**33**	*α*-Funebrene	1391	1492	-	-	4.5
**34**	Sativene	1400	1537	-	0.6	-
**35**	*α*-Bourbonene	1402	1521	1.7	-	-
**36**	*β*-Caryophyllene ^c^	1419	1593	6.6	-	1.9
**37**	*β*-Cedrene	1422	1613	-	-	8.9
**38**	*α*-Bergamotene	1426	1543	-	-	2.1
**39**	*γ*-Elemene	1434	1627	0.5	-	-
**40**	*α*-Cedrene	1443	1587	-	1.0	0.5
**41**	*α*-Humulene ^c^	1452	1608	3.1	-	2.5
**42**	Isocaryophyllene	1461	1655	0.1	-	-
**43**	Dehydroaromadendrene	1462	1621	-	-	3.0
**44**	*γ*-Muurolene	1471	1684	0.5	-	-
**45**	*Allo*-aromadendrene	1482	1638	0.8	-	-
**46**	*α*-Curcumene	1486	1767	-	0.5	-
**47**	*Allo*-aromadendr-9-ene	1496	1677	-	-	1.0
**48**	*β*-Bisabolene ^c^	1509	1720	26.8	-	-
**49**	*γ*-Cadinene	1511	1766	5.5	-	-
**50**	*δ*-Cadinene	1525	1751	-	-	1.8
**51**	*α*-Bisabolene	1530	1762	0.4	-	-
**52**	9,10-Dehydro-Isolongifolene	1531	1759	-	-	2.3
**53**	*β*-Sesquiphellandrene	1534	1754	0.3	0.5	-
**54**	*trans*-Nerolidol ^c^	1554	2029	0.2	-	-
**55**	Ledol	1565	2033	0.4	-	-
**56**	Caryophyllene oxide ^c^	1571	1999	3.1	-	2.8
**57**	Palustrol	1578	1931	0.2	-	-
**58**	Globulol	1580	2078	2.0	-	4.5
**59**	Spathulenol ^c^	1584	2100	0.5	-	1.5
**60**	*τ*-Cadinol	1613	2167	0.4	-	-
**61**	Humulane-1,6-dien-3-ol	1616	2009	2.2	-	-
**62**	Epiglobulol	1629	2008	0.2	0.9	13.5
**63**	Longifolenaldehyde	1635	1944	-	-	14.5
**64**	Cubenol	1639	2055	-	1.2	2.8
**65**	*δ*-Cadinol	1641	2179	-	-	1.4
**66**	Ledene oxide	1646	2021	-	-	12.1
**67**	*α*-Cadinol	1653	2222	0.3	t	-
**68**	*β*-Bisabolol	1678	2196	-	-	3.9
**69**	8-Cedren-13-ol	1690	2380	-	-	5.7
**70**	*α*-Bisabolol	1707	2224	1.5	-	-
**71**	6,10,14-Trimethyl-2-pentadecanone	1830	2127	-	-	0.3
**72**	Corymbolone	1918	2355	-	-	0.5
**73**	Manool ^c^	2036	2488	0.3	-	-
**74**	Phytol ^c^	2125	2587	0.2	-	0.4
	Monoterpene hydrocarbons			27.0	72.7	0.9
	Oxygenated monoterpenes			0.3	12.2	-
	Sesquiterpene hydrocarbons			48.3	2.6	30.8
	Oxygenated sesquiterpenes			11.0	2.1	63.5
	Others			4.6	3.0	0.7
	Total			91.2	92.6	95.9

LRI ^a^: Linear retention index on a DB-5; LRI ^b^: Linear retention index on a HP-Innowax column; ^c^: Co-elution with authentic sample.

**Table 2 molecules-26-00643-t002:** Main compounds (>3%) of the essential oils obtained by hydrodistillation from *T. montanum*, *T. capitatum* and *T. flavum* reported in the literature.

Section *Polium*			*T. montanum*						
Origin	Abr	Parts	Compounds	MH	OM	SH	OS	O	Ref.
Croatia	***A***(***m***)	ap	germacrene D (17.2), *β*-pinene (12.9), *β*-caryophyllene (7.1), limonene (4.6), myrcene (4.2), linalool (3.6), *β*-bourbonene (3.4), hexacosane (3.4), pentacosane (3.3), tetracosane (3.1)	24.4	12.4	35.1	5.1	22.5	[25]
Turkey	***B***(***m***)	ap	sabinene (11.3), *δ*-cadinene (6.3), germacrene D (5.8), *α*-copaene (5.7), (*E*)-*β*-farnesene (5.5), *τ*-cadinol (5.4), *α*-pinene (5.2), linalool (3.2), *β*-pinene (3.1)	26.5	9.0	33.5	13.1	8.8	[26]
Serbia-Mont Mt. Orjen	***C***(***m***)	ap	germacrene D (15.0), *α*-pinene (12.4), *β*-eudesmol (l0.l), *β*-caryophyllene (6.9), *β*-pinene (4.8), *δ*-cadinene (4.5), *γ*-cadinene (4.1), cadinol (3.6), bicyclogermacrene (3.5)	21.2	1.8	48.6	24.1	1.4	[27]
Serbia, Jadovnik	***D***(***m***)	ap	*δ*-cadinene (17.2), *β*-selinene (8.2), *α*-calacorene (5.0), cadalene (4.9), caryophyllene (4.3), copaene (4.2), torreyol (3.9), terpine-4-ol (3.9), cadina-1,4-diene (3.4), *β*-sesquiphellandrene (3.3), *γ*-curcumene (3.2), *τ*-cadinol (3.1)	1.1	6.5	74.2	7.0	9.2	[28,29]
Serbia, Jabura	***E***(***m***)	ap	*δ*-cadinene (8.1), *β*-caryophyllene (5.1), *τ*-muurolol (4.2), *α*-pinene (4.0), dehydrosesquicineole (3.9), *γ*-cadinene (3.6), *α*-cadinol (3.5)	7.9	14.1	39.3	33.4	3.7	[30]
**Section *Polium***			***T. capitatum***						
Algeria, Boussaada	***A***(***c***)	ap	*τ*-cadinol (18.3), germacrene D (15.3), *β*-pinene (10.5), carvacrol (5.5), bicyclogermacrene (5.5), *α*-pinene (4.1), limonene (3.1)	22.0	12.4	31.2	25.5	0	[31]
Algeria, Bouira	***B***(***c***)	ap	germacrene D (25.0), *β*-pinene (11.3), bicyclogermacrene (10.4), spathulenol (5.8), limonene (4.0), *τ*-cadinol (3.5)	23.1	1.3	43.7	13.5	0.1	[32]
Bulgaria	***C***(***c***)	ap	*β*-pinene (26.8), germacrene D (17.7), *α*-pinene (9.3), limonene (6.4), *E*-nerodilol (4.6), bicyclogermacrene (4.0), myrtenal (3.3), spathulenol (3.2)	45.7	17.2	27.7	5.8	0	[33]
Corsica, Bustanico	***D***(***c***)	ap	*α*-pinene (24.1), *β*-pinene (9.2), *α*-thujene (8.1), terpinen-4-ol (6.2), limonene (5.2), sabinene (4.1), *p*-cymene (4.0)	59.3	29.6	4.4	4.8	0.8	[34,35]
Corsica, Corte	***E***(***c***)	ap	*α*-pinene (28.8), *β*-pinene (7.2), *p*-cymene (7.0), *α*-thujene (5.0), terpinen-4-ol (4.6), *p*-cymene-4-ol (3.0), limonene (3.0)	56.2	18.5	1.5	2.9	0.4	[36]
Greece, Crete	***F***(***c***)	ap	caryophyllene (9.8), carvacrol (10.1), torreyol (7.6), *α*-cadinol (4.5), *cis*-verbenone (3.7), germacrene D (3.1), *α*-humulene (3.8), *δ*-cadinene (3.1), *E*-nerolidol (3.0)	1.2	13.6	32.7	23.1	22.9	[37]
Greece, Athens	***G***(***c***)	ap	*α*-pinene (14.8), *β*-pinene (12.8), *β*-caryophyllene (11.3), *τ*-cadinol (7.7), myrcene (5.5), germacrene D (4.8), sabinene (4.7), *α*-humulene (3.3), *α*-cadinol (3.2), limonene (3.1)	44.2	9.2	25.1	21.2	0	[38]
Greece, Kos	***H***(***c***)	lv + fl	germacrene D (53.7), (*E*)-*β*-farnesene (10.0), bicyclogermacrene (9.1), spathulenol (3.2), limonene (3.1)	5.5	0	79.9	3.2	0	[39]
Greece, Crete	***I***(***c***)	ap	caryophyllene (10.1), carvacrol (9.6), torreyol (6.5), caryophyllene oxide (5.0), *α*-cadinol (4.0), *cis*-verbenone (4.0) germacrene D (3.9), *α*-humulene (3.4), *δ*-cadinene (3.1), germacrene D-4-ol (3.0)	1.2	15.0	33.9	21.3	21.0	[40]
Iran	***L***(***c***)	ap	*α*-cadinol (46.2), caryophyllene oxide (25.9), *epi*-*α*-muurolol (8.1), cadalene (3.7)	0.1	2.5	3.7	92.4	1.3	[41]
Morocco	***M***(***c***)	ap	*endo*-borneol (33.0), *δ*-cadinene (19.6), bornyle acetate (15.6), *α*-terpineol (12.0), *α*-thujol (10.9)	0	74.1	20.6	0.3	3.7	[42]
Portugal, Fonte Coberta	***N***(***c***)	ap	*τ*-cadinol (5.5), *p*-menthan-3-one (7.7), *β*-caryophyllene (4.8), germacrene D (3.5), terpine-4-ol (3.5), *α*-cadinol (3.2), *δ*-cadinene (3.0)	11.2	33.0	24.3	14.9	0.4	[43]
Portugal, Rabaçal	***O***(***c***)	ap	sabinene (11.2), *β*-pinene (10.3), *α*-pinene (7.7), *δ*-cadinene (4.9), *β*-caryophyllene (3.8), germacrene D (3.6), myrcene (3.5)	43.9	24.9	23.2	7.5	1.2	[43]
Portugal, Cantanhede	***P***(***c***)	ap	*τ*-cadinol (24.1), *α*-cadinol (9.8), *γ*-cadinene (5.5), *δ*-cadinene (3.7), *δ*-verbenol (3.5), *β*-caryophyllene (3.3), caryophyllene oxide (3.1)	7.3	19.8	23.0	39.7	0.4	[43]
Portugal, Cantanhede	***Q***(***c***)	ap	*δ*-cadinene (7.5), *β*-caryophyllene (5.4), *α*-cadinol (4.2), germacrene D (3.5), *τ*-cadinol (3.2), terpine-4-ol (3.0)	7.6	19.7	32.2	23.4	0.8	[43]
Portugal, Serra D′Aire	***R***(***c***)	ap	*δ*-cadinene (9.8), *β*-pinene (5.0), *β*-caryophyllene (4.6), *α*-cadinol (4.6), sabinene (3.1), *τ*-cadinol (3.0)	17.0	12.5	33.2	20.4	1.0	[43]
Sardinia, Porticciolo	***S***(***c***)	ap	limonene (20.6), *α*-pinene (20.4), *E*-nerolidol (16.7), *β*-pinene (7.6), myrcene (7.5)	62.8	11.4	3.8	16.7	0	[44]
Sardinia, Gennargentu	***T***(***c***)	ap	limonene (17.2), *α*-pinene (12.5), *α*-*trans*-bergamotene (12.2), humulene epoxide II (9.2), *δ*-cadinene (7.7), *β*-calacorene (5.6), *β*-pinene (4.5)	40.2	4.1	37.8	10.0	0	[44]
Serbia	***U***(***c***)	ap	germacrene D (31.8), linalool (14.0), *β*-pinene (10.7), *β*-caryophyllene (8.8), bicyclogermacrene (6.2), *α*-pinene (3.5)	20.2	17.3	55.1	4.5	0.2	[43]
**Section *Chamaedrys***			***T. flavum***						
Croatia, Dalmatia	***A***(***f***)	ap	*α*-pinene (17.3), *β*-caryophyllene (15.8), *β*-pinene (11.2), *allo*-aromadendrene (9.2), limonene + 1,8-cineole (6.2), *α*-cubebene (4.3), *γ*-terpinene (3.5), *δ*-cadinene (3.2)	42.5	7.0	43.4	4.3	1.8	[45]
Croatia, Split	***B***(***f***)	ap	*β*-caryophyllene (23.1), germacrene-D (15.3), *α*-pinene (10.5), *β*-pinene (8.4), limonene (7.9), *n*-amyl-isovalerate (3.7)	28.4	3.4	50.8	5.0	9.7	[25]
Greece, Mt Pileum	***C***(***f***)	ap	caryophyllene (12.2), 4-vinyl-guaicol (9.7), caryophyllene oxide (7.9), *α*-humulene (6.0), linalool (3.4), *β*-bourbonene (3.1)	1.3	9.1	39.0	15.5	26.8	[37]
Iran	***D***(***f***)	lv	*β*-caryophyllene (30.6), germacrene-D (21.3), *α*-humulene (8.4), *τ*-cadinol (6.9), *δ*-cadinene (4.9), *trans*-*α*-bergamotene (4.8), spathulenol (4.5), caryophyllene oxide (3.8), *β*-bisabolene (3.1)	0	0	82.2	17.7	0	[46]
Montenegro, River Moraca	***E***(***f***)	ap	*β*-bisabolene (35.0), *α*-pinene (17.5), *β*-pinene (11.5), limonene (6.4), *β*-caryophyllene (5.4), *α*-humulene (3.6)	38.8	4.4	50.1	1.5	2.9	[47]
Italy, Marche	***F***(***f***)	ap, dry	(*Z*,*E*)-*α*-farnesene (11.5), linalool (7.6), *β*-bisabolene (7.5), (*E*)-*β*-farnesene (7.3), 11-*αH*-himachal-4-en-1-*β*-ol (6.2), *β*-caryophyllene (5.7), germacrene D (5.5), *α*-pinene (5.3), *β*-pinene (4.5), limonene (3.5)	14.2	8.9	48.5	11.1	16.6	[47]
Italy, Marche	***G***(***f***)	ap, fresh	(*Z*,*E*)-*α*-farnesene (14.9), 11-*αH*-himachal-4-en-1-*β*-ol (10.1), germacrene D (6.6), (*E*)-*β*-farnesene (5.7), *β*-caryophyllene (5.1), *β*-bisabolene (5.0)	11.5	9.4	49.4	12.2	16,5	[47]
Italy, Liguria	***H***(***f***)	ap	*α*-pinene (19.0), germacrene D (11.9), *β*-pinene (10.6), limonene (9.0), *α*-bulnesene (8.9), (*Z*,*E*)-farnesolo (4.7), (*E*)-*β*-farnesene (3.3)	46.5	1.2	35.6	7.0	5.8	[48]
Italy, Tuscany	***I***(***f***)	ap	*α*-pinene (22.6), *β*-pinene (15.8), limonene (13.2), germacrene D (6.9), (*E*,*E*)-*α*-farnesene (4.8)	58.5	2.7	19.1	3.6	6.0	[48]
Greece, Zakynthos	***L***(***f***)	ap	caryophyllene (13.5), caryophyllene oxide (8.5), 4-vinyl-guaiacol (6.0), *α*-humulene (5.0), hexahydrofarnesyl acetone (3.4), *α*-copaene (3.3)	0.5	7.5	41.3	14.4	26.0	[49]
Corsica	***M***(***f***)	ap	*α*-pinene (21.9), limonene (20.0), *β*-pinene (18.1), (*Z*)-*α*-ocimene (15.5)	80.7	6.7	2.3	0.8	4.9	[35]
Tunisia	***N***(***f***)	ap	*β*-caryophyllene (32.5), *α*-humulene (17.8), germacrene D (6.0), caryophyllene oxide (4.9), (*Z*)-*γ*-bisabolene (4.0)	0.2	2.0	68.0	11.9	11.1	[50]

ap = aerial parts; fl = flowers; lv = leaves. MH = monoterpene hydrocarbons; OM = oxygenated monoterpenes; SH = sesquiterpene hydrocarbons; OS = oxygenated sesquiterpenes; O = others.

## Data Availability

Data sharing is not applicable to this article.

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
