# Peer review of "The Essential Oil Compositions of Three Teucrium Taxa Growing Wild in Sicily: HCA and PCA Analyses"

_molecules, 2021, doi:10.3390/molecules26030643_

Round 1
Reviewer 1 Report
In the Introduction section, after consulting the Plant List [1]; I did not find the data reported by the authors for the species belonging to the genus Teucrium. Moreover, after the genus Teucrium it is not added by L..
In the Introduction section, Scordium must be in italic.
From lines 81-97, it is necessary to add the references from which the authors describe the two species of Teucrium.
At the end on the Introduction, it should be clear the main objective of the work and not in the middle of the Introduction. Such means that the Introduction should be reformulated.
In Table 1, what is the meaning of O(f), V(c), F(m)? The question is done because along the text until to this Table there is no information about the meaning of the abbreviations. Only after Table 2 and Figures it is possible to understand the meaning, nevertheless such should be clearer and not intuitive.
In the Results and Discussion:
Table 2 reports the main components of the essential oils, obtained by hydrodistillation, of the other populations of T. flavum, previously investigated, as well as of the other accessions of T. montanum and T. capitatum collected in different countries. A comparison of our data with those reported in literature (Table 2).
It is not clear how did the authors make the selection of references for doing the compilation and statistical analyses. It would be interesting to write something about this previous selection of data.
Lines 214-215: SO or OS (63.5%)? The others two oils, Serbia Jadovnik (D(m)) and Sicily (F(m)), for the high content of SH (74.2%) and SO (63.5%) respectively, were considered as two separate classes, without any similarity with the others EOs.
Author Response
In the Introduction section, after consulting the Plant List [1]; I did not find the data reported by the authors for the species belonging to the genus Teucrium. Moreover, after the genus Teucrium it is not added by L..
The sentence has been revised(Lines 36-38)
L. was deleted
In the Introduction section, Scordium must be in italic.
Corrected
From lines 81-97, it is necessary to add the references from which the authors describe the two species of Teucrium.
Reference inserted
At the end on the Introduction, it should be clear the main objective of the work and not in the middle of the Introduction. Such means that the Introduction should be reformulated.
Introduction has been reformulated (lines 94-103)
In Table 1, what is the meaning of O(f), V(c), F(m)? The question is done because along the text until to this Table there is no information about the meaning of the abbreviations. Only after Table 2 and Figures it is possible to understand the meaning, nevertheless such should be clearer and not intuitive.
The abbreviations O(f), V(c), F(m) have been inserted along the text before Table 2
In the Results and Discussion:
Table 2 reports the main components of the essential oils, obtained by hydrodistillation, of the other populations of T. flavum, previously investigated, as well as of the other accessions of T. montanum and T. capitatum collected in different countries. A comparison of our data with those reported in literature (Table 2).
It is not clear how did the authors make the selection of references for doing the compilation and statistical analyses. It would be interesting to write something about this previous selection of data.
A phrase in order to clarify has been inserted in the text. (lines 125-127)
Lines 214-215: SO or OS (63.5%)? The others two oils, Serbia Jadovnik (D(m)) and Sicily (F(m)), for the high content of SH (74.2%) and SO (63.5%) respectively, were considered as two separate classes, without any similarity with the others EOs.
It was a misprint. Corrected
Reviewer 2 Report
The submitted manuscript described EO of Teucrium in Sicily and geographical variation using HCA and PCA. It is recommendable for publication.
1. Introduction
The authors performed HCA and PCA analysis. However, no objection of this performance was not appealed. Please indicate it in introduction section.
2. GC Column:
GC-MS: HP-5MS
GC-FID: DB-5
The two column use the same stationary phase. That means the eluting order (RI) of components is almost same. However in Table 1. LRIa and LRIb are quite different.
The values of LRIb is close to that obtained from polar column.
Please re-check the column used in Shimadzu GC.
2. Fig. 1-3. Describe about distance (----2).
Although the authors described HCA analysis in title,
there is no description on HCA analysis results.
What distance between and in clusters?
It would like to indicate using dendrogram.
3. Line 98-105: This sentence dealt with non-volatile components of Teucrium spp. It seem no relationship with this study. It would be better to be deleted.
4. Minor
Line 14: change Perillen to Perillene.
Line 243. Fig 1. should be Fig 3.
Do not put space between number and %.
Line 293. Indicate the company of Wely275, NIST 17, ADAMS, FFNSC2 libraries.
Line 294. only Cluster Analysis (CA) methods was described briefly. Describe about PCA analysis.
Author Response
The submitted manuscript described EO of Teucrium in Sicily and geographical variation using HCA and PCA. It is recommendable for publication.
- Introduction
The authors performed HCA and PCA analysis. However, no objection of this performance was not appealed. Please indicate it in introduction section.
Introduction has been reformulated
- GC Column:
GC-MS: HP-5MS
GC-FID: DB-5
The two column use the same stationary phase. That means the eluting order (RI) of components is almost same. However in Table 1. LRIa and LRIb are quite different.
The values of LRIb is close to that obtained from polar column.
Please re-check the column used in Shimadzu GC.
The correct columns have been inserted in the text.
- Fig. 1-3. Describe about distance (----2).
Although the authors described HCA analysis in title,
there is no description on HCA analysis results.
What distance between and in clusters?
It would like to indicate using dendrogram.
The dendrograms have been inserted (Fig. 2, Fig. 4 and Fig.6), marking the distance between the clusters. An accurate description of HCA analysis has been reported for every single species.
- Line 98-105: This sentence dealt with non-volatile components of Teucrium spp. It seem no relationship with this study. It would be better to be deleted.
This part has been deleted and also its references
- Minor
Line 14: change Perillen to Perillene. Corrected
Line 243. Fig 1. should be Fig 3. Corrected
Do not put space between number and %. Corrected
Line 293. Indicate the company of Wely275, NIST 17, ADAMS, FFNSC2 libraries.
Companies were added
Line 294. only Cluster Analysis (CA) methods was described briefly. Describe about PCA analysis.
Improved
Reviewer 3 Report
Dear Authors,
this work is focused on the characterization of the essential oils of 3 species belonging to the genus Teucrium. The expansion of knowledge relating to the rich biodiversity present in the Mediterranean basin is certainly very important and interesting. In your work, however, I am not very clear about your hypotheses. For what purpose did you do this work? It seems to me that the premises are missing towards the end of the Introduction Section. An HCA is inserted in the title, which I do not find in the text. Please go into these aspects further.
Here are my other indications:
- line 21: the nomenclator is missing.
- lines 29-31: first in full and then acronym in parentheses.
- from line 69 to 97: I believe they should be mixed in the M&M together with the other information relating to the collection habitat. The variations in the chemical composition of the oils also derives a lot from the conditions of growth and development.
- line 122: Corsica (France).
- Table 1 and 2: they must be inserted after their citation in the text and not distant.
- Not all the codes in the figure are specified in the captions of the figures.
- There are two Fig. 1.
- line 157: the title of the paragraph is too vague. In this paragraph many sources of literature are also indicated but then in the text there is very little discussion. Please expand.
Author Response
Dear Authors,
this work is focused on the characterization of the essential oils of 3 species belonging to the genus Teucrium. The expansion of knowledge relating to the rich biodiversity present in the Mediterranean basin is certainly very important and interesting. In your work, however, I am not very clear about your hypotheses. For what purpose did you do this work? It seems to me that the premises are missing towards the end of the Introduction Section. An HCA is inserted in the title, which I do not find in the text. Please go into these aspects further.
Introduction has been reformulated
Here are my other indications:
- line 21: the nomenclator is missing. Inserted
- lines 29-31: first in full and then acronym in parentheses. Corrected
- from line 69 to 97: I believe they should be mixed in the M&M together with the other information relating to the collection habitat. The variations in the chemical composition of the oils also derives a lot from the conditions of growth and development.
Habitat information have been added
- line 122: Corsica (France). Inserted
- Table 1 and 2: they must be inserted after their citation in the text and not distant.
Tables 1 and 2 have been inserted after their citation
- Not all the codes in the figure are specified in the captions of the figures.
A note citing Table 2 has been inserted
- There are two Fig. 1. Corrected
- line 157: the title of the paragraph is too vague. In this paragraph many sources of literature are also indicated but then in the text there is very little discussion. Please expand.
Modified
Round 2
Reviewer 1 Report
The authors have responded to all questions.
Author Response
The authors have responded to all questions.
Reviewer 3 Report
Dear Authors, the work has been significantly improved. I still have a specific to do: - lines 169-172 go to M&M. Specifically, it is not clear how you came to the decision to indicate the similarity value 2 as the cut off and then the value 1. Since the description of the statistical results are the fundamental part of this work, I ask the authors to better define this part . After that, for me the work is publishable.
- in the caption of Figure 2 the reference codes to the samples are missing.
Author Response
Dear Authors, the work has been significantly improved. I still have a specific to do: - lines 169-172 go to M&M.
Lines have been moved to M&M
Specifically, it is not clear how you came to the decision to indicate the similarity value 2 as the cut off and then the value 1. Since the description of the statistical results are the fundamental part of this work, I ask the authors to better define this part . After that, for me the work is publishable.
A paragraph has been inserted in M&M.
- in the caption of Figure 2 the reference codes to the samples are missing.
Captions have been inserted in Figures 2, 4 and 6